# Frequency stabilization and noise-induced spectral narrowing in resonators with zero dispersion

L. Huang [1,2], S.M. Soskin[3,4], I.A. Khovanov [5], R. Mannella [6], K. Ninios[7] & H.B. Chan[1,2]

Mechanical resonators are widely used as precision clocks and sensitive detectors that rely on the stability of their eigenfrequencies. The phase noise is determined by different factors including thermal noise, frequency noise of the resonator and noise in the feedback circuitry. Increasing the vibration amplitude can mitigate some of these effects but the improvements are limited by nonlinearities that are particularly strong for miniaturized micro- and nano-mechanical systems. Here we design a micromechanical resonator with non-monotonic dependence of the eigenfrequency on energy. Near the extremum, where the dispersion of the eigenfrequency is zero, the system regains certain characteristics of a linear resonator, albeit at large amplitudes. The spectral peak undergoes narrowing when the noise intensity is increased. With the resonator serving as the frequency-selecting element in a feedback loop, the phase noise at the extremum amplitude is ~3 times smaller than the minimal noise in the conventional nonlinear regime.

[1] Department of Physics, The Hong Kong University of Science and Technology, Clear Water Bay, Kowloon, Hong Kong, China. [2] William Mong Institute of Nano Science and Technology, The Hong Kong University of Science and Technology, Clear Water Bay, Kowloon, Hong Kong, China. [3] Institute of Semiconductor Physics, National Academy of Sciences of Ukraine, 03028 Kiev, Ukraine. [4] Department of Physics, Lancaster University, Lancaster LA1 4YB, UK. [5] School of Engineering, University of Warwick, Coventry CV4 7AL, UK. [6] Dipartimento di Fisica, Universita di Pisa, 56127 Pisa, Italy. [7] Department of Physics, University of Florida, Gainesville, FL 32611, USA. Correspondence and requests for materials should be addressed to H.B.C. (email: hochan@ust.hk)

Resonant detectors and frequency standards typically operate in the linear regime, where the eigenfrequency $\omega$ is independent of the amplitude and hence the energy. The frequency stability depends on the width of the spectral peak, which is in turn proportional to the damping constant. For systems coupled to a thermal bath, the damping constant is related to thermal fluctuations via the fluctuation–dissipation theorem. The spectral width is an important parameter in mesoscopic resonators, including nanomechanical devices[1–6], Josephson junctions[7,8], and optomechanical systems[9]. Apart from practical implications on the phase noise, the spectral width also provides useful information on the underlying relaxation mechanisms[10–12]. Recently, there has been much interest in different factors that modify the spectral response of mesoscopic resonators. It has been observed in nanomechanical resonators that spectral broadening can arise from the fluctuations of the eigenfrequency[13–16]. Alternatively, the interplay between nonlinearity, fluctuations, and relaxation has also been found to have a profound effect on the spectral lineshape[17,18].

Mechanical oscillators used in sensing or timekeeping often rely on a feedback circuit to maintain self-sustained oscillations[19–22] so that the eigenfrequency of the resonator can be tracked. A number of factors contribute to the phase noise, including the spectral width of the resonator and noise generated by the feedback circuit. These effects can be mitigated by increasing the vibration amplitude, provided that resonator remains in the linear regime. The improvement in frequency stability, however, is limited by nonlinear effects. When the vibration amplitude increases beyond the linear regime, the eigenfrequency is no longer independent of energy. The nonlinearity converts amplitude fluctuations into frequency fluctuations, leading to degradations in phase stability[21]. Such nonlinear effects are particularly strong in miniaturized silicon micro- and nanomechanical devices that have become viable candidates to replace bulky quartz resonators commonly used as frequency references[23]. Stabilization of the frequency in the presence of strong nonlinearity has attracted much interest recently. For example, one approach makes use of higher-order modes of the resonator. At amplitudes where the two modes are in internal resonance[21,24–26], the frequency fluctuations of the lower mode are shown to be significantly reduced[21].

On a more fundamental level, the nonlinearity modifies the spectral lineshape of resonators. When resonators are subjected to thermal and/or external fluctuations, the noise creates a distribution of energies that, together with the nonlinearity, tends to broaden the peak in the power spectrum[18,27]. The overall shape of the spectral peak is determined by the interplay of the frequency straggling due to nonlinearity and the frequency uncertainty associated with decay. As the noise intensity increases, the peak broadens, generally, and becomes asymmetric. Changes in spectral peaks with increasing temperature were observed in the optical spectra of localized and resonant vibrations in solids[28]. However, a full quantitative comparison with theory was difficult due to uncertainties in system parameters and difficulties in controlling them.

In this work, we design an electromechanical resonator with non-monotonic dependence of the eigenfrequency $\omega$ on energy $E$ to yield improved spectral width and frequency stability. At the extremum energy $E_{zd}$, the dispersion of $\omega(E)$ is zero so that $d\omega/dE = 0$. Our system opens the possibility for the investigation of a wealth of phenomena in "zero-dispersion" systems[29–40] that are associated with the large contribution of vibrations over a range of energies in the vicinity of $E_{zd}$ occurring at almost the same frequency. These phenomena bear resemblance to van Hove singularities in the density of states of solids[41,42]. In our mechanical resonator, the non-monotonic dependence of $\omega$ on

energy $E$ is achieved by tuning the negative nonlinearity induced by electrostatic force from a nearby electrode relative to the positive intrinsic nonlinearity of the springs. We demonstrate that in the zero-dispersion regime, the peak of the fluctuation spectrum undergoes substantial narrowing as the noise intensity is increased, in good agreement with theory. We also set our devices into self-sustained oscillations with active feedback. The standard deviation of the oscillation frequency at the optimal vibration amplitude is demonstrated to be a factor of ~3 smaller in the zero-dispersion regime compared to the minimal noise in the conventional nonlinear regime. To our knowledge, zero-dispersion phenomena have not been directly observed in physical systems other than analog circuits and computer simulations[30,32]. Our findings demonstrate that zero-dispersion phenomena can play an important role in micro/nanomechanical systems, potentially leading to new methods of detection and frequency stabilization.

## Results

**Micromechanical trampoline resonator.** Figure 1a shows the micromechanical trampoline resonator used in our experiment. The device consists of a highly doped polycrystalline silicon plate ($3.5\,\mu m \times 300\,\mu m \times 300\,\mu m$) suspended by springs at its four corners (Fig. 1b). Electrical voltage $V$ is applied to a fixed electrode of the same size underneath the top plate to excite vibrations of the top plate in the fundamental mode that involves translational motion normal to the substrate. Measurements are performed at 4 K and $10^{-6}$ Torr. The top plate can be modeled as a resonator in one dimension:

$$\ddot{q} = -2\Gamma\dot{q} + \frac{1}{m}[F_s(q) + F_e(q)], \tag{1}$$

where $q$ is the plate displacement, $m$ is the mass, and $\Gamma$ (0.496 rad s$^{-1}$) is the damping constant. $F_s$ is the restoring force of the spring:

$$F_s(q) \approx -m\omega_s^2 q - m\beta_s q^3 - m\mu_s q^5. \tag{2}$$

The spring nonlinearity coefficients $\beta_s$ and $\mu_s$ are positive because of the increase in tension with displacement $q$. Asymmetry of the spring is negligible at relevant scales of $q$. $F_e(q)$ is the electrostatic force:

$$F_e(q) \equiv F_e(q, V^2) = \frac{1}{2}\frac{dC(q)}{dq}V^2, \tag{3}$$

where $C = \varepsilon S/(g - q)$ is the capacitance between the plate and the electrode, $\varepsilon$ is the permittivity of free space, $S$ is the area of the plate, and $g$ is the initial gap of $2\,\mu m$ between the plate and the electrode.

The applied voltage $V$ between the plate and the electrode includes up to four components:

$$V = V_{dc} + V_d(t) + V_n(t) + V_{c1}(t), \tag{4}$$

where $V_{dc}$ is the dc voltage. $V_d(t) = V_{ac}\cos(\omega_d t)$ (with $V_{ac} \ll |V_{dc}|$) is the ac driving at frequency $\omega_d$ close to the eigenfrequency of the fundamental mode. It leads to a resonant periodic electrostatic force on the plate. $V_n(t)$ is the noise voltage that generates a random force. The last term $V_{c1}(t) = V_c\cos(\omega_c t)$ represents a sinusoidal carrier voltage at amplitude $V_c = 300$ mV and frequency $\omega_c = 2.5 \times 10^7$ rad s$^{-1}$. It is used to measure the change in capacitance between the plate and the electrode, from which $q$ can be inferred (see Methods). For all results shown in this paper, $V_{ac} \ll V_c \ll |V_{dc}|$. Besides, since $\omega_c \gg \omega_d$, the only effect of the carrier voltage on the dynamics of $q$ is a slight increase of the effective value of the dc voltage $\tilde{V}_{dc}^2 = V_{dc}^2 + V_c^2/2$.

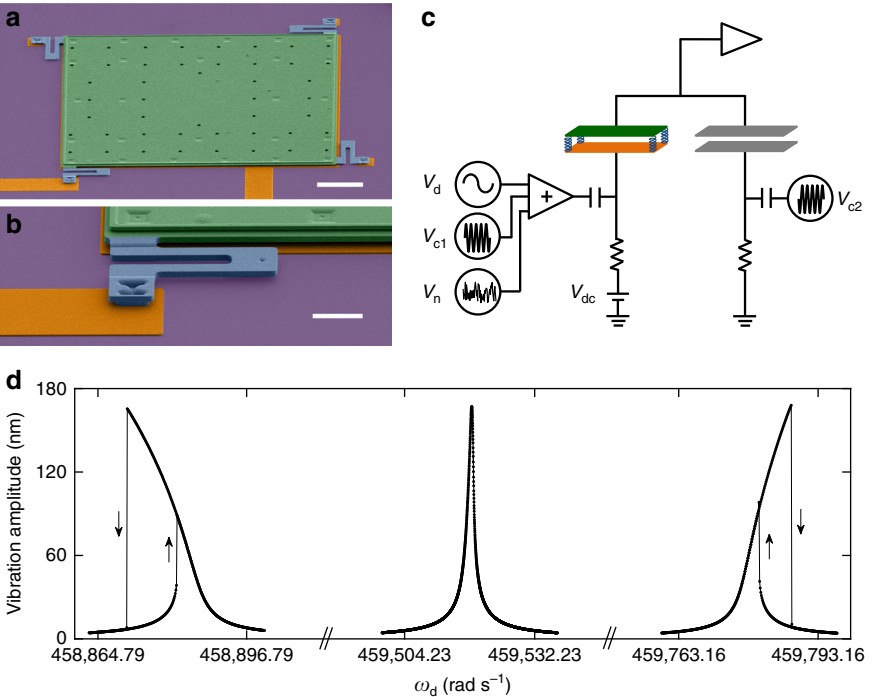

**Fig. 1** Micromechanical trampoline resonator. **a** Colorized scanning electron micrograph of the resonator that consists of a movable silicon top plate (green) supported by serpentine springs (blue) at its four corners. The white scale bar measures 50 µm. **b** A close-up on one of the four springs at a corner. The scale bar measures 15 µm. **c** Simplified schematic showing the electrostatic actuation and detection of vibrations of the top plate using a capacitance bridge. **d** The electrostatic contribution to the Duffing nonlinearity can be controlled by $V_{dc}$. Measured vibration amplitude is plotted as a function of driving frequency for a fixed driving force amplitude and three different $V_{dc}$: −2.3, −1.6, and −1.2 V from left to right, where the Duffing nonlinearity changes from positive to near zero to negative

**Dependence of frequency of eigenoscillations on amplitude**. One of the most important characteristics of the resonator is the dependence of the frequency of eigenoscillation of the resonator $\omega$ on its amplitude $a$: it determines characteristic features of dynamics both in case of resonant periodic driving[43,44] and in case of driving by noise[27,38]. The dependence of the electrostatic force on $V_{dc}$ allows us to control $\omega(a)$ and turn the zero-dispersion behavior on or off. Such control is particularly efficient in our resonator because the gap $g$ is much smaller than the characteristic distance scale of nonlinearity in the spring restoring force $L_n = \omega_s/\sqrt{\beta_s}$. Apart from the efficient control, it also allows us to develop for $\omega(a)$ an asymptotic theory with a small parameter $\tilde{\nu}$ that is proportional to $\tilde{V}_{dc}^2$:

$$\tilde{\nu} \equiv \tilde{\nu}(\tilde{V}_{dc}^2) = \frac{\varepsilon S}{2mg^3\omega_s^2}\tilde{V}_{dc}^2. \tag{5}$$

As it will be shown below, the range of $\tilde{V}_{dc}^2$ relevant to our experiment is

$$\tilde{V}_{dc}^2 \sim \beta_s mg^5/(2\varepsilon S). \tag{6}$$

Therefore, $\tilde{\nu}$ in this range is of the order of $[g/(2L_n)]^2$, which is $\ll 1$. As we will later explain, $\tilde{\nu}$ plays a crucial role in determining whether the resonator exhibits zero-dispersion.

At the equilibrium position $q_{eq} \approx \tilde{\nu}g \ll g$, the spring and electrostatic forces balance each other. We consider the force $F = F_s + F_e$ as a function of $x \equiv q - q_{eq}$ and expand $F(x)$ into Taylor series up to the fifth order (see Supplementary Note 1):

$$F(x) \equiv F(x, \tilde{V}_{dc}^2) = m\sum_{n=1}^{5}\alpha_n x^n, \tag{7}$$

where the odd coefficients contain contributions from both the spring and the electrostatic forces:

$$\alpha_1 \approx \omega_s^2(1 - 2\tilde{\nu}), \tag{8}$$

$$\alpha_3 \approx \beta_s - \frac{4\omega_s^2}{g^2}\tilde{\nu} \equiv \beta_s\left\{1 - \frac{\tilde{\nu}}{[g/(2L_n)]^2}\right\}, \tag{9}$$

$$\alpha_5 \approx \mu_s - \frac{6\omega_s^2}{g^4}\tilde{\nu} \equiv -\frac{6\omega_s^2}{g^4}\tilde{\nu}\left\{1 - \frac{[g/(2L_n)]^2}{\tilde{\nu}}\frac{8\mu_s L_n^2}{3\beta_s}\left(\frac{g}{2L_n}\right)^2\right\} \tag{10}$$

(relatively small terms proportional to $\tilde{\nu}^n$ with $n \geq 2$ are neglected in Eqs. (8–10)). In contrast, the even coefficients contain no contributions from the spring [see Eq. (2)]. They originate exclusively from the electrostatic force and, to the lowest order in $\tilde{\nu}$, they are proportional to $\tilde{\nu}\omega_s^2$, namely, $\alpha_2 \approx 3\tilde{\nu}\omega_s^2/g$ and $\alpha_4 \approx 5\tilde{\nu}\omega_s^2/g^3$.

Most nonlinear micro- and nanomechanical resonant systems can be modeled as Duffing oscillators that includes up to the cubic nonlinear term. Obtaining zero-dispersion phenomena in our system requires keeping nonlinear terms up to at least the fifth order. They generate a fourth-order term in the dependence of $\omega$ on the vibration amplitude $a$, in addition to the usual quadratic term as in a Duffing oscillator (Supplementary Note 1):

$$\omega \approx \omega_1 + \kappa a^2 + \eta a^4, \tag{11}$$

where

$$\omega_1^2 \approx \omega_s^2(1 - 2\tilde{\nu}), \tag{12}$$

$$\kappa \approx \frac{3\beta_s}{8\omega_s}\left\{1 - \frac{\tilde{\nu}}{[g/(2L_n)]^2}\right\}, \tag{13}$$

$$\eta \approx -\frac{15\omega_s \tilde{\nu}}{8g^4}. \tag{14}$$

In contrast to $\eta$ (Eq. (14)) which is necessarily negative, $\kappa$ (Eq. (13)) can be chosen to be either positive or negative by controlling $\tilde{\nu}$ via $\tilde{V}_{dc}^2$ (Supplementary Note 1). When the signs of $\kappa$ and $\eta$ are opposite, the eigenfrequency possesses a parabolic maximum as a function of $a^2$, providing a versatile platform for investigating zero-dispersion phenomena.

**Resonance response curves**. We study the dependence of the vibration amplitude on the frequency of a sinusoidal drive when $V_n(t)$ is set to zero. The equation of motion can thus be simplified to:

$$\ddot{x} + 2\Gamma\dot{x} + \omega_1^2 x + \alpha_3 x^3 + \alpha_5 x^5 = (F_{ac}/m)\cos(\omega_d t), \tag{15}$$

where $F_{ac} = \frac{\epsilon S}{g^2}|V_{dc}|V_{ac}$. Contributions from $\alpha_i$ with even $i$ are negligible (see Supplementary Note 1). The results for small $F_{ac}$ are plotted as resonance response curves in Fig. 1d. For small $F_{ac}$, the vibration amplitude $A$ is also small such that the term $\alpha_5 x^5$ in Eq. (15) and the term $\eta a^4$ in Eq. (11) for $a = A$ can be neglected. The system reduces to the periodically driven underdamped Duffing oscillator then. Figure 1d shows the electrostatic spring softening effect: the resonance peaks shift to lower frequency as $V_{dc}^2$ increases, as described by Eq. (12). Apart from the frequency shift, the nonlinear coefficients also change with $V_{dc}^2$. At small $V_{dc}^2$, the inherently positive spring contribution to the cubic nonlinearity $\alpha_3$ strongly dominates over the negative contribution from the electrostatic force, so that the resonance curve bends towards high frequencies and exhibits clockwise hysteresis[45,46]. While the spring contribution is fixed, the absolute value of the negative electrostatic contribution increases in proportion to $\tilde{V}_{dc}^2$. At $V_{dc} = -1.6$ V, it balances the spring contribution so that $\alpha_3 \sim 0$. As a result, the asymmetry in the resonance peak is largely gone and the peak is well fitted by the response of a damped harmonic oscillator. As $V_{dc}^2$ further increases, $\alpha_3$ becomes negative so that the resonance peak bends towards low frequencies. A somewhat similar evolution of the resonance response curves was observed in refs. [47,48], but the reverse of the curve bending was caused there primarily by the strong growth of the $\alpha_2$ contribution, in contrast to our case. Regardless of the origin of such an evolution of resonance curves, the regime in which $\kappa$ becomes zero may be used to increase the dynamic range of micromechanical devices[48].

Figure 2a shows measured and calculated (see Supplementary Note 1 for details) resonance curves at $V_{dc} = -2.3$ V, where the nonlinearity due to the electrostatic force dominates over the spring nonlinearity, for driving amplitudes of 36.7 and 202 pN. As the driving amplitude increases, the hysteresis loop widens and the frequency $\omega_{peak}$ with the largest vibration amplitude $A_{peak}$ shifts further towards low frequencies. By measuring the resonance curves at many values of the driving amplitude and identifying the peak vibration amplitude, we obtain the backbone line $A_{peak}(\omega_{peak})$. The reversed function $\omega_{peak}(A_{peak})$ provides a good approximation of the frequency of eigenoscillations (see Supplementary Note 1) with amplitude $a$ equal to $A_{peak}$[43]. Figure 2b plots the dependence of $\omega_{peak}$ on $A_{peak}^2$, which we call the "transformed backbone line." The solid line represents a fit using Eq. (11) with $a = A_{peak}$, yielding $\omega_1 = \omega_{1a} = 458,874.45$ rad s$^{-1}$, $\kappa = -2.16 \times 10^{14}$ rad s$^{-1}$ m$^{-2}$ and $\eta = -1.13 \times 10^{26}$ rad s$^{-1}$ m$^{-4}$.

If $V_{dc} > -1.6$ V, the transformed backbone line becomes non-monotonic, that is, the device acquires zero-dispersion properties. Figure 2c shows resonance curves for two driving amplitudes at $V_{dc} = -1.51$ V. At the lower driving amplitude, the curve bends towards higher frequencies due to the positive $\kappa$. The sign of $\kappa$ is changed compared to Fig. 2a because, with a smaller $V_{dc}^2$, the spring contribution dominates over the electrostatic force. In Eq. (13), the ratio $\tilde{\nu}/[g/(2L_n)]^2$ becomes slightly smaller than 1. At larger driving amplitude, the curve acquires a mixed behavior: it bends to higher frequencies at sufficiently low vibration amplitude $A$ but, as $A$ exceeds certain limit, the bending changes direction towards lower frequencies since the negative fourth power term in $\omega(A)$ [Eq. (11)] overcomes the positive quadratic term (Supplementary Note 1 explains why $\eta$ is negative in our experiment). There exists two separate frequency ranges with hysteretic behavior[49]. The system can have one, two, or three co-existing stable vibration states depending on the excitation frequency, with a total of four bifurcation frequencies. Figure 2d shows (by circles) the measured non-monotonic transformed backbone line $\omega_{peak}(A_{peak}^2)$. The solid line represents a fit using Eq. (11) with $a = A_{peak}$, yielding $\omega_1 = \omega_{1b} = 459,572.9$ rad s$^{-1}$, $\kappa = 3.77 \times 10^{13}$ rad s$^{-1}$ m$^{-2}$, and $\eta = -7.37 \times 10^{25}$ rad s$^{-1}$ m$^{-4}$. Unlike Fig. 2b, $\kappa$ and $\eta$ have opposite signs. The transformed backbone line exhibits a local maximum, at which $d\omega/dE = 0$, that provides a platform for exploring and utilizing zero-dispersion phenomena.

**Noise-induced spectral widening and narrowing**. Next, we investigate the spectral density of fluctuations of our resonator in the conventional Duffing and the zero-dispersion regimes. The measurements are performed by injecting noise $V_n$ in the driving voltage [Eq. (4)] while turning off the ac drive $V_d$. Noise $V_n$ is Gaussian, centered around $\omega_1$ with a bandwidth of ~5000 rad s$^{-1}$ for the conventional nonlinear regime and ~2000 rad s$^{-1}$ for the zero-dispersion regime (see Methods). All the resonance and spectral peaks investigated (such as those in Fig. 1) lie well within this bandwidth, so that the noise can be assumed white in the context of the fluctuation spectrum, playing a role similar to thermo-mechanical noise (Supplementary Note 2). The dynamics is well described by Eq. (15) with the following modifications: (i) the ac force $F_{ac}\cos(\omega_d t)$ being replaced by the auxiliary white noise $F_n^{(w)}$ with intensity $D_w$ defined in Supplementary Eq. (38) and (ii) $\tilde{V}_{dc}^2$ in the left-hand side being replaced by the slightly larger value as described in Supplementary Note 2.

The spectral density of fluctuations $\tilde{Q}(\omega)$ is defined as the half-Fourier transform of the correlation function of coordinate $Q(t)$[29–32,36,38,50]:

$$Q(t) \equiv \langle (q(t) - \langle q \rangle)(q(0) - \langle q \rangle)\rangle, \tag{16}$$

$$\tilde{Q}(\omega) \equiv \frac{1}{\pi}\text{Re}\left[\int_0^\infty dt\, Q(t)\exp(-i\omega t)\right]. \tag{17}$$

Details of the methods of measurements of $\tilde{Q}(\omega)$ and of its theoretical calculations are given in Supplementary Note 3. The procedure involves transformation from the description in terms of dynamical variables to that of the non-stationary conditional probability density and, accordingly, from the description by means of the Langevin equation to that by means of the Fokker–Planck equation (FPE)[50], with the further substantial simplification of the solution of the FPE and calculation of the spectrum using the method suggested in ref. [30].

Figure 3a, c plot the measured spectral density of fluctuations (see Methods) at $V_{dc}$ of $-2.3$ and $-1.51$ V, corresponding to the monotonic and non-monotonic transformed backbone lines in Fig. 2b, d, respectively. When the noise intensity $D_w$ is small, the spectrum is Lorentzian as in a harmonic oscillator. The peak is symmetric, centered at the natural frequency, and the width is

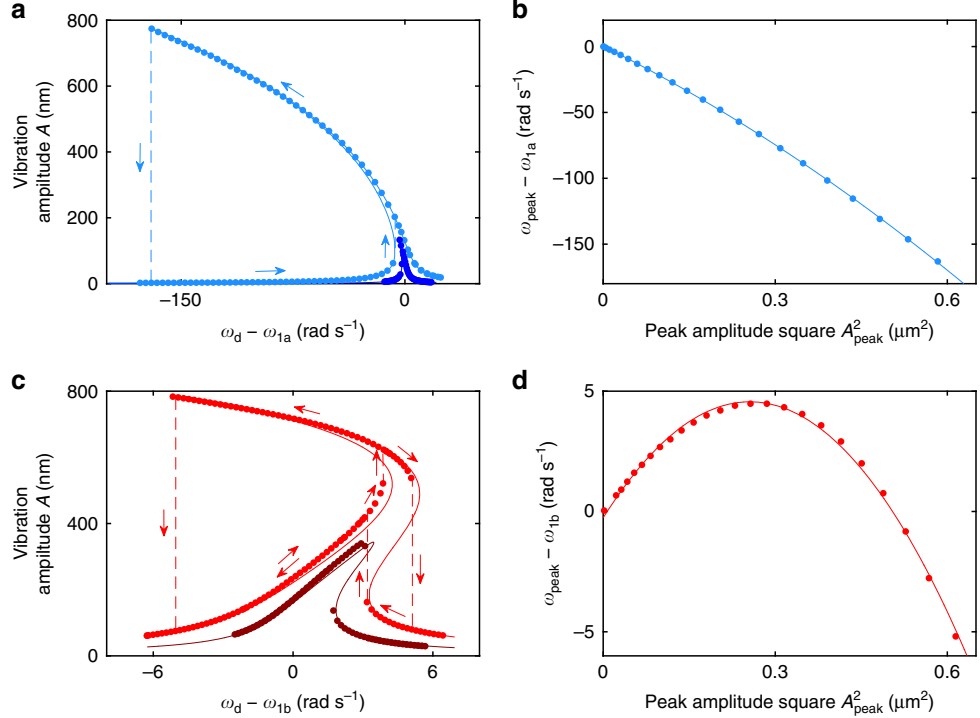

**Fig. 2** Resonance curves and transformed backbone lines in the conventional nonlinear and zero-dispersion regimes. **a** At $V_{dc} = -2.3$ V, the resonator is in the conventional nonlinear regime. The resonance curve bends towards lower frequencies as the driving amplitude $F_{ac}$ is increased from 36.7 pN (dark blue) to 202 pN (blue). Measurement and theory (Supplementary Note 1) are represented by circles and lines, respectively. **b** Measured transformed backbone line (circles) showing the shift in $\omega_{peak}$ relative to the natural frequency versus the squared vibration amplitude at the peak of the resonance curve as the driving amplitude increases. The solid line is a quadratic fit. **c** At $V_{dc} = -1.51$ V, the resonator is in the zero-dispersion regime. The resonance curve for $F_{ac} = 75.2$ pN (dark red) bends to higher frequencies. For $F_{ac} = 171$ pN (red), the initial bending is also towards higher frequencies, but as the vibration amplitude further increases, the bending changes direction. **d** Accordingly, the transformed backbone line becomes non-monotonic and exhibits a maximum. The error bars are smaller than the dot size in all panels

solely determined by the damping constant $\Gamma$. As $D_w$ increases, the interplay between fluctuations and nonlinearity leads to profound changes in the width, height, and shape of the peaks. The noise-induced energy straggling leads to a corresponding frequency straggling. Therefore, the width of the spectral peak no longer depends solely on the damping constant. As shown in Fig. 3a for the case where the transformed backbone line is monotonic, the width of the peak increases with $D_w$ and the peaks become asymmetric. Figure 3b shows that the measured change in the width of the peak is roughly proportional to the noise intensity. In Fig. 3a, b, the solid lines show results of the theoretical calculations with no fitting parameters. Given that the asymptotic theory is only of the lowest order, the agreement may be considered satisfactory.

When the system is in the zero-dispersion regime at $V_{dc} = -1.51$ V, there exists an energy $E_{zd}$ (with corresponding eigenfrequency $\omega_{zd} \equiv \omega(E_{zd})$) at which $d\omega/dE = 0$. In the limit when damping approaches zero, the spectral density of fluctuation is predicted to diverge at $\omega_{zd}$[29] because vibrations in a relatively large range of energies take place with almost the same frequency. In our system, the ratio $(\omega_{zd} - \omega_1)/\Gamma$ is not sufficiently large to distinctly reveal this peak. Nevertheless, the reduced frequency straggling associated with $d\omega/dE = 0$ results in a decrease of the spectral width as the noise intensity is increased. Such noise-induced spectral narrowing is clearly seen in Fig. 3c, where the spectral peak at $D_w = 6840$ pN$^2$ Hz$^{-1}$ has a smaller width compared to $D_w = 2028$ pN$^2$ Hz$^{-1}$. Figure 3d plots the dependence of the spectral width on noise intensity. As $D_w$ increases, the width initially increases, attaining a maximum value at $D_w \sim 2030$ pN$^2$ Hz$^{-1}$. Upon further increase in $D_w$, it drops by

about 30% at $D_w \sim 6800$ pN$^2$ Hz$^{-1}$. At even larger $D_w$, the spectral width resumes its increase. Similar to the conventional case, the theoretically calculated spectra demonstrate a reasonable agreement with the experiments with no fitting parameters.

**Frequency stabilization in the zero-dispersion regime**. The reduced spectral width in the zero-dispersion regime offers new opportunities for frequency stabilization in applications of frequency standards. In the linear regime, the spectral width directly determines the frequency stability when the resonator is driven into self-sustained oscillations via active feedback, where the mechanical resonator serves as the frequency-selecting element[20,21]. To evade thermal noise or noise from the feedback amplifier, it is beneficial to maximize the vibration amplitude of the mechanical resonator, provided that the response of the mechanical element remains linear. In practice, however, as the vibration amplitude increases, nonlinear effects become more prominent. Fluctuations in the vibration amplitude are converted via the nonlinearity into frequency fluctuations. Increasing the vibration amplitude eventually leads to a degradation of phase stability of self-sustained oscillations. To demonstrate this phenomenon, we drive our resonator into self-sustained vibrations using a phase-locked loop (see Methods). We then record the frequency and amplitude of this oscillator as a function of time, using an averaging time of ~160 ms. Figure 4b plots the distributions of the measured frequency and amplitude in the conventional nonlinear case as the purple circles for different driving amplitudes of the feedback loop. The phase delay $\Delta\varphi$ between the oscillations and the drive of the phase-locked loop is adjusted for

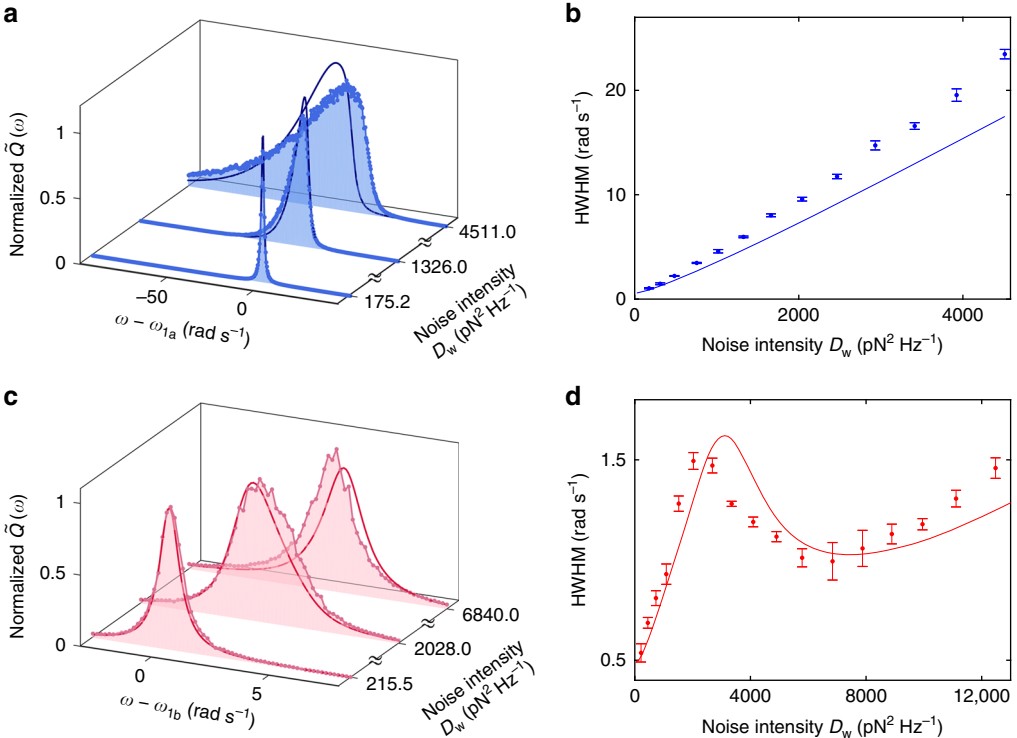

**Fig. 3** Spectral widening/narrowing in the conventional/zero-dispersion case. **a** In the conventional nonlinear regime ($V_{dc} = -2.3$ V), the spectral peak becomes wider and asymmetric as the noise intensity is increased. Each peak is normalized by its maximum measured value. **b** Measured (circles) and calculated (solid line) half-width at half-maximum (HWHM) of the spectral peaks increase monotonically with the noise intensity. **c**, **d** Similar plots for the zero-dispersion regime for $V_{dc} = -1.51$ V. Note the difference of scales as compared to **a**, **b**, respectively. The HWHM shows a distinctly non-monotonic dependence on the noise intensity, yielding a range of noise intensity where the spectral peak becomes narrower as the noise intensity increases. Error bars represent ±1 s.e.

maximum vibration amplitude, so that the center of the distributions coincides with the peaks of the corresponding resonance response curves measured when the driving frequency is swept with the feedback turned off (blue circles). The width of the distributions along the frequency axis, which is related to the standard deviation $\sigma_\omega$ of the frequency, display a non-monotonic dependence on the amplitude of the active drive. As shown in Fig. 4b for a monotonic backbone curve, a stronger drive initially improves the phase stability for small oscillation amplitudes. The frequency standard deviation $\sigma_\omega$ decreases when the driving amplitude $F_{ac}$ is increased from 9.5 to 63.35 pN. However, at large vibration amplitudes, the nonlinear effects convert amplitude noise into phase noise, leading to an increase in $\sigma_\omega$ when the driving amplitude is increased to 158.4 pN. The blue curve in Fig. 4d plots $\sigma_\omega$ versus the driving amplitude of the feedback loop for the case when the transformed backbone line is monotonic, showing an optimal driving amplitude that yields the minimum standard deviation: $\sigma_\omega^{(min)} \approx 0.106$ rad s$^{-1}$ at $F_{ac} \approx 63$ pN. Here, the data are taken in a separate cool down compared to Figs. 1–3. The noise intensity $D_w$ is fixed at 6800 pN$^2$ Hz$^{-1}$ for self-sustained oscillations and set to zero for the resonance response curves without feedback.

The frequency stability can be significantly improved when $V_{dc}$ is adjusted to yield zero-dispersion behavior. Figure 4c shows measurements for the case of a non-monotonic backbone line using four driving amplitudes (22.2, 71.3, 182, and 269 pN). For force vibrations without feedback, the driving amplitude of 182 pN (yielding the third resonance curve from bottom) marks the transition of the resonance curves from bending towards high frequency to the mixed behavior. For self-sustained oscillations with feedback, the distribution of measured frequencies (plotted in

purple) shows a remarkable narrowing for this driving amplitude as the system approaches the zero-dispersion regime, where the eigenfrequency of the resonator is locally independent of the vibration amplitude. In Fig. 4d, the dependence of $\sigma_\omega$ on driving amplitude for this case is plotted in red, showing that the initial drop of $\sigma_\omega$ with driving amplitude largely coincides with the blue curve for a nonlinear resonator with a monotonic transformed backbone line. As the driving amplitude continues to increase beyond 60 pN, the two oscillators behave differently. Instead of a sharp increase with the driving amplitude, $\sigma_\omega$ for the resonator with non-monotonic backbone exhibits only a slight increase followed by a gentle drop. The drop becomes steeper as the driving amplitude further increases. $\sigma_\omega$ eventually attains the absolute minimum $\sigma_{\omega,zd}^{(min)} \approx 0.034$ rad s$^{-1}$ at driving amplitude of 182 pN (same as the driving amplitude for the second highest resonance response curve in Fig. 4c), beyond which $\sigma_\omega$ increases sharply. Remarkably, the minimal $\sigma_\omega$ for the zero-dispersion regime at 182 pN is about 3 times smaller compared to the minimal $\sigma_\omega$ in the conventional regime. Further improvements appear feasible if the zero-dispersion energy can be pushed higher.

In analyzing the frequency stability of oscillators and clocks, the Allan deviation is often used to eliminate systematic errors such as long-term frequency drifts. For the results shown in Fig. 4d, the frequency of the phase-locked loop is recorded for only 120 s for each driving amplitude. Over such a short duration, the effect of frequency drift is negligible in our system. In Supplementary Note 4, we show that the improvement of the minimal Allan deviation in the zero-dispersion regime over its counterpart in the conventional regime is about 3.6 for averaging time of 1 s, close to the improvement in $\sigma_\omega$ in Fig. 4d.

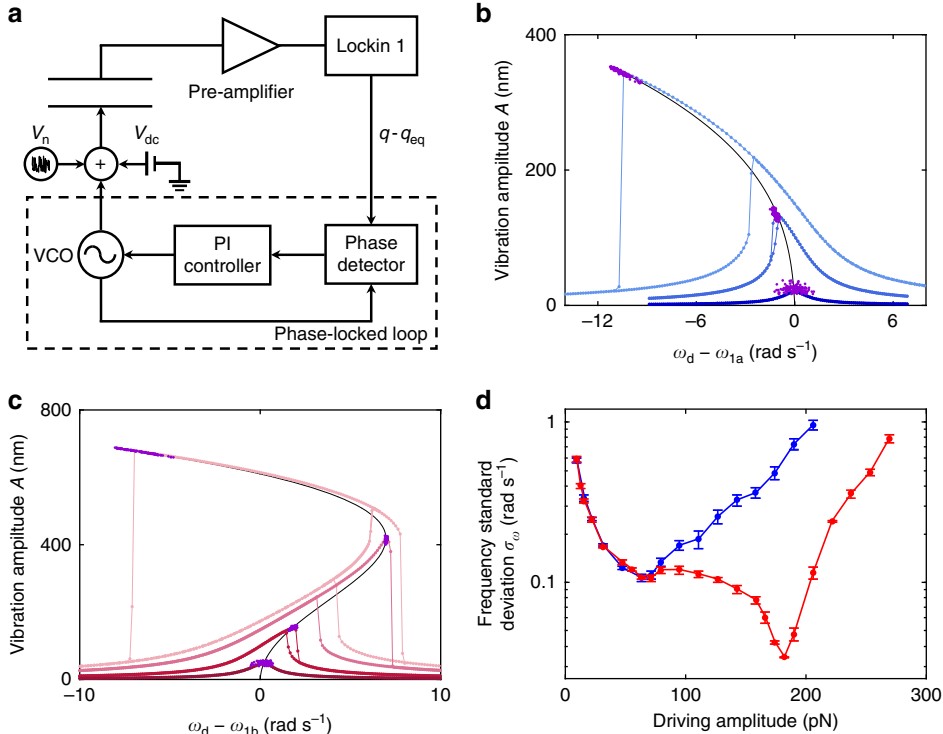

**Fig. 4** Frequency stabilization in the zero-dispersion regime compared to the conventional regime. **a** Schematic for driving the resonator into self-sustained oscillations with a phase-locked loop. **b** Forced and self-sustained oscillations in the conventional nonlinear regime. With no feedback, the forced vibration amplitude as a function of driving frequency is measured at driving amplitudes of (bottom to top) 9.5, 63.35, and 158.4 pN. The solid black line represents the backbone line. With the feedback turned on and the phase delay chosen to maximize the self-oscillation amplitude, the measured distributions of the self-sustained oscillation amplitude and frequency are plotted in purple. **c** Similar plot for the zero-dispersion regime. From bottom to top, the driving amplitudes are 22.2, 71.3, 182, and 269 pN. The stabilization of frequency in the zero-dispersion regime is demonstrated in the third curve at driving amplitude of 182 pN. **d** Standard deviation of the frequency of self-sustained oscillations $\sigma_\omega$ versus driving amplitude for oscillators in the conventional regime (blue) and the zero-dispersion regime (red). The zero-dispersion regime yields the minimum standard deviation $\sigma_\omega^{(min)}$ about a factor of ~3 smaller than that of the conventional regime. Error bars represent ±1 s.e.

## Discussion

Our findings show that the spectral width of micromechanical resonators is determined by the interplay of damping, nonlinearity, and additive noise. In a conventional nonlinear oscillator, where the transformed backbone line is monotonic, the fluctuation spectrum peak widens as the noise intensity increases. For a resonator with zero dispersion, where the transformed backbone line is non-monotonic, the spectral peak undergoes narrowing as the noise intensity is increased in some range related to $E_{zd}$, in agreement with theoretical calculations.

For the zero-dispersion resonator, the optimal frequency stability of self-sustained oscillations shows a considerable improvement as compared to conventional nonlinear resonator. One advantage of such an approach to stabilize the frequency in the presence of strong nonlinearity is that only a single vibrational mode is involved. Matching the frequency of other modes[21] is therefore unnecessary, potentially simplifying the operation.

The zero dispersion in our resonator is obtained by tuning the negative electrostatic nonlinearity relative to the positive intrinsic nonlinearity of the springs. Non-monotonicity of a transformed backbone line can also be achieved using other methods, such as the coupling of two nanomechanical resonators[51] or the negative nonlinear friction induced by dynamical backaction from a photon or phonon cavity[52]. Thus, zero dispersion is a rather generic feature of mechanical resonators and such resonators provide a well-controlled platform to investigate and exploit zero-dispersion phenomena reported in the present work as well as others that have not yet been experimentally observed.

The improvement of frequency stability for resonators driven into self-sustained vibrations using feedback was demonstrated for applied noise that is additive. Such noise plays a role similar to thermal–mechanical noise that leads to both phase and amplitude fluctuations. At small vibration amplitudes, the direct contribution to phase fluctuations dominates. As the amplitude increases, conventional nonlinearity converts amplitude noise into phase noise. The effect of such a conversion is substantially reduced in the zero-dispersion regime with $d\omega/dE = 0$ so that amplitude fluctuations do not affect the phase stability even at large vibration amplitudes. If, instead, the phase noise is dominated by fluctuations in the eigenfrequency of the resonator[13–16], operating the feedback loop in the regime of zero dispersion does not yield improvement in phase stability because amplitude fluctuations are not involved.

The main effect of the injected noise in our experiment is that it increases the effective temperature of the resonator. If the injected noise is removed, the thermal noise at temperature of 4 K leads to random motion of the plate. In our experiment, such thermal motion is too small to be resolved due to noise in the detection circuit at room temperature. To estimate the effect of zero dispersion on preventing the conversion of amplitude fluctuations originating from thermal noise into phase fluctuations, we perform numerical simulations assuming that no additional noise is introduced by the feedback loop and the detection circuit. The optimal $\sigma_\omega$ in the zero-dispersion regime is found to be smaller than the conventional nonlinear regime by a factor of 4, comparable to the case measured in our experiment with injected noise.

Apart from the applied noise and thermal–mechanical noise, the feedback circuit also contributes to noise in the phase of oscillations. In particular, we consider the effects from the output of the voltage-controlled oscillator (VCO) that is used to drive the mechanical resonator. Fluctuations in the VCO output amplitude leads to amplitude noise in vibrations, but the phase dynamics are not directly affected if the resonator is linear. However, when the vibration amplitude is large, nonlinearity becomes strong and such amplitude noise could be converted into phase fluctuations. Designing and operating the resonator in the zero-dispersion regime reduces such conversion and is therefore expected to yield improved frequency stability, similar to the case of additive noise reported here. The experimental demonstration of such improvement, however, is outside the scope of this paper.

For the non-monotonic backbone curves, the optimal phase stability for self-sustained oscillations is achieved at the vibration amplitude $A_{zd}$ at which $d\omega/dE = 0$. By eliminating the extra contributions to phase noise converted from amplitude noise, the phase stability becomes comparable to an oscillator undergoing purely linear vibrations at amplitude $A_{zd}$. It is well known that for linear resonators, the frequency standard deviation for self-sustained vibrations is inversely proportional to the vibration amplitude[53]. Therefore, the optimal frequency standard deviation for an oscillator in the zero-dispersion regime is expected to follow a similar dependence on $A_{zd}$, that is, $\sigma_{\omega,zd}^{(min)} \propto 1/A_{zd}$. Ideally, the phase stability can be improved by choosing $A_{zd}$ to be as large as possible. The upper limit in our experiment is given by the gap $g$ between the movable plate and the fixed electrode. In practice, operating close to this upper limit could lead to the plate touching the electrode when occasional large fluctuations occur. In Figs. 3, 4, $A_{zd}$ is ~400 nm, about a factor of 5 smaller than $g$. Assuming that $A_{zd}$ can be increased to 80% of the upper limit, the ratio of the optimal frequency standard deviation in the zero-dispersion regime to that in the conventional regime could reach ~12, a factor of ~4 larger than in Fig. 4d. Demonstrating such best-case scenario in our device, however, is not the main goal of this paper. Instead, we choose $A_{zd}$ to be smaller than $g$ to show the general features of zero-dispersion phenomena. For example, to reveal the local minimum of the spectral width as a function of the noise intensity in Fig. 3d, it is necessary to go beyond the zero-dispersion regime using vibration amplitude that approaches the maximum displacement. Promising future research directions for phase stabilization using zero dispersion include new approaches to yield large $A_{zd}$, as well as the effect on the phase stability due to other phenomena that emerges when the vibration amplitude becomes large, such as nonlinear friction.

Finally, we note that zero-dispersion phenomena often become more prominent when the ratio $(\omega_{zd} - \omega_1)/\Gamma$ increases. In the current experiment, this ratio is ~10. If the ratio can be significantly increased in future designs, a wealth of zero-dispersion phenomena is expected to occur. Examples include the appearance of extremely sharp peaks in fluctuation spectra[29,32,38] and the onset of deterministic chaos at extraordinarily low periodic drive amplitudes[37,38,54], to name just a few. Further studies of these and other zero-dispersion phenomena may not only be of fundamental interest, but could also lead to new methods of signal detection and other applications.

During the preparation of this manuscript, the authors learned that the theory for reduction of phase noise in self-sustained oscillations exploiting the zero-dispersion property is being developed by Miller[55]. These results could allow for further improvements in phase stability using zero-dispersion phenomena.

## Methods

**Detection scheme**. Displacement of the top plate is inferred from the capacitance change $\Delta C$ between the top plate and the fixed electrode underneath. As shown in Fig. 1c, two ac voltages $V_{c1}$ and $V_{c2}$ at the same frequency $f_c = \omega_c/2\pi$ (4 MHz), with comparable amplitude (300 mV) and opposite phases, are applied to the bottom electrode and a standard capacitor, respectively. The other plate of the standard capacitor and the top plate of the resonator are electrically connected. Displacement of the top plate leads to a change in the capacitance between the top plate and the fixed electrode. Therefore, motion of the top plate modulates the amplitude of ac signal at the carrier frequency on the top plate. This ac signal is measured with a lock-in amplifier referenced to $f_c$. Changes in the lock-in output are proportional to the deviation of the displacement $q$ from the equilibrium position $q_{eq}$.

**Noise generation**. The noise voltage is generated from the Johnson noise of a 50 $\Omega$ resistor at room temperature. After the Johnson noise is amplified and passed through a band-pass filter with center frequency $f_{center}$ and bandwidth $f_{bd}$, it is mixed with a noise carrier voltage at frequency 50 kHz, creating two sidebands centered at 50 kHz $\pm f_{center}$.

In the experiments on the fluctuation spectra measurement, $f_{center} = 23,034$ Hz and $f_{bd} = 741$ Hz for the conventional nonlinear regime ($V_{dc} = -2.3$ V), while $f_{center} = 23,143$ Hz and $f_{bd} = 298$ Hz for the zero-dispersion regime ($V_{dc} = -1.51$ V). In the experiments on the frequency stabilization, $f_{center} = 23,237$ Hz and $f_{bd} = 748$ Hz for the conventional nonlinear regime ($V_{dc} = -1.863$ V), while $f_{center} = 23,271$ Hz and $f_{bd} = 749$ Hz for the zero-dispersion regime ($V_{dc} = -1.59$ V). The eigenfrequency of the resonator lies within the upper sideband. Its bandwidth is much larger than the width of the spectral peaks discussed in the main text.

**Measurement of the spectral density of fluctuations**. When we measure the spectral density of fluctuations, the periodic drive is turned off. $V_n(t)$ produces a random force that plays a similar role as thermo-mechanical noise. The first lock-in amplifier produces an output that is proportional to $q$. A second lock-in amplifier, referenced at frequency $\omega_1/2\pi$ and with a bandwidth of 100 Hz, yields amplitudes of oscillations $X(t)$ and $Y(t)$ that are in phase and out of phase with the reference:

$$q(t) = X(t)\cos(\omega_1 t) + Y(t)\sin(\omega_1 t). \tag{18}$$

The spectral density of fluctuations is calculated then as follows:

$$\tilde{Q}(\omega) = \frac{1}{N}\sum_\tau\sum_t\left\{[X(t+\tau)+iY(t+\tau)][X(t)-iY(t)]e^{-i(\omega-\omega_d)\tau}\right\}, \tag{19}$$

where the discretization number $N$ is 4000 and 4044 in the conventional and zero-dispersion cases, respectively. The discretization step (identical for $\tau$ and $t$) is 4.44 and 8.93 ms in the conventional and zero-dispersion cases, respectively.

**Oscillations sustained by active feedback**. Lock-in amplifier 1 produces a signal $V_{out1}(t)$ that is proportional to the deviation of the plate displacement $q(t)$ from the equilibrium position $q_{eq}$. Self-sustained oscillations are maintained by feedback using the phase-locked loop of lock-in amplifier 2. Specifically, $V_{out1}(t)$ is fed into the reference of lock-in amplifier 2. The phase-locked loop adjusts the frequency $\omega_d$ of the periodic drive to maintain a fixed, controllable phase delay $\Delta\varphi$ between $V_{out1}(t)$ and the periodic drive. $\Delta\varphi$ is adjusted to maximize the oscillation amplitude. Data in Fig. 4 are taken in a separate cool down from Figs. 1–3 so that the device parameters are slightly changed. For the conventional regime (Fig. 4b), $V_{dc}$ is chosen to be $-1.863$ V, giving $\omega_1 = \omega_{1a} \equiv 460,180.7$ rad s$^{-1}$ and a transformed backbone line with $\kappa = -6.19 \times 10^{13}$ rad s$^{-1}$ m$^{-2}$ and $\eta = -2.23 \times 10^{26}$ rad s$^{-1}$ m$^{-4}$. For the zero-dispersion regime (Fig. 4c), $V_{dc}$ is $-1.59$ V, giving $\omega_1 = \omega_{1b} \equiv 460,376.2$ rad s$^{-1}$ and a transformed backbone line with $\kappa = 6.12 \times 10^{13}$ rad s$^{-1}$ m$^{-2}$ and $\eta = -1.39 \times 10^{26}$ rad s$^{-1}$ m$^{-4}$. $\kappa$ for the two different $V_{dc}$ have almost the same magnitude but opposite sign.

## Data availability
The data that support the findings of this study are available from the corresponding author on request.

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

## Acknowledgements

L.H. and H.B.C. are supported by Research Grant Council of HKSAR, China (Project No. 16303215). K.N. is supported by NSF No. DMR- 0645448. S.M.S. acknowledges the support by Volkswagen Foundation (Grant No. 90418), the support of his visits to Pisa by Pisa University and to University of Warwick by the Institute of Advanced Studies and School of Engineering of University of Warwick, and the access to the e-library of Lancaster University provided by his Honorary Visiting Researcher position there. We acknowledge discussions during the preparation of the manuscript with M.I. Dykman and S. Shaw on the theory for reduction of phase noise in self-sustained oscillations exploiting the zero-dispersion property.

## Author contributions

H.B.C. conceived the idea of the work and designed the experiments. S.M.S., I.A.K., and R.M. developed the theory. L.H. and K.N. performed the experiments and analyzed the data. L.H., H.B.C., and S.M.S. co-wrote the paper.

## Additional information

**Competing interests:** The authors declare no competing interests.

**Peer Review Information:** *Nature Communications* thanks the anonymous reviewers for their contribution to the peer review of this work. Peer reviewer reports are available.

