## [Peer Review File · Nature Communications]

Reviewers' comments:

Reviewer #1 (Remarks to the Author):

This manuscript reports the suppression of resonance peak broadening and the reduction of frequency fluctuation induced by mechanical nonlinearity. Authors fabricated a device which enables the electrical control of the 3rd and 5th order nonlinearity to construct a "zero-dispersion" region and confirmed the spectral narrowing and improved frequency stability. The concept is quite new and the demonstration is nearly first time (except for Ref 30). The stability improvement of mechanical resonator is extremely important in its applications to sensors and timing devices so that I believe that the result includes enough new results to warrant the publication in Nature Communications. I have only small minor comments to allow the publication.

1. Noise is directly applied to the device shown in Fig.4 in the experimental demonstration of frequency stabilization. However, I am not sure if the similar effect can be expected in the case when the noise was generated in the loop circuit because the fluctuation in VCO amplitude also increases the noise amplitude. If I correctly understand the result, the reduction of frequency deviation was observed because the injected noise is independent from the operation of upper stage of the circuit for example the pre-amplifier? I imagine the injected noise is also increased when the noise is amplified by the loop gain.

2. Please estimate the effect expected in the case of Brownian motion.

3. Which is the major effect of applied noise, amplitude or phase? How is the effect of this method against intrinsic phase noise in mechanical resonator (say, mass fluctuation etc.)?

4. The zero-dispersion amplitude is 400 nm in this experiment. It would be ideal if the amplitude can be increased as much as possible, but it is not the case in reality. Comment on the upper limit.

Reviewer #3 (Remarks to the Author):

I have read the manuscript "Frequency stabilization and noise-induced spectral narrowing in resonators with zero dispersion" with interest. In response to your specific questions –

1. Major claims? The most important claims are for the first experimental observation of the phenomenon of noise-induced spectral narrowing in a nonlinear oscillator that has a nonmonotonic dependence of eigenfrequency on oscillation energy, together with other phenomena associated with the presence of a zero-dispersion point. These ideas first arose some three decades ago, when some of the phenomena in question were seen in an analogue simulator, giving rise to rapid development of the fluctuational dynamics theory needed to account for the observations. The micromechanical silicon resonator developed by the present authors is more complicated than the earlier analogue model, and displays a richer range of behaviours. Yet the same basic ideas still apply, and the theory has been successfully developed to account for the measurements – which is the other major claim of the manuscript.

2. Novelty and interest? The central theoretical ideas about what happens near a zero-dispersion point have been around since the 1990s and were extensively reviewed in Ref 38 (2003); but what is now reported is the first experimental realisation and the extension of the theory to encompass the (inevitably) greater complexity exhibited by reality. So, yes, the claims are definitely novel and, in my opinion, interesting. I should add that I am not myself a theorist, so I cannot vouch for the correctness of all the theoretical details, but what is written is all valid so far as I can tell. Given that the authors effectively provide a "recipe" for both building and understanding micro- or nano-mechanical resonators with zero-dispersion properties, and that there are potential applications related e.g. to frequency stabilisation, I anticipate interest within the relevant community.

3. Interest within a wider field? Given that nonlinear oscillators abound in practically all areas of science, I believe that the claims will also be of broader interest going far beyond specialists in nano/micromechanical resonators.

4. Likely influence on thinking in the field? Given its novelty, widespread potential applicability, and possible early/medium-term technological applications – as well as its closing a conceptual loop of some 30 years' standing – I would expect the paper to exert significant influence.

The manuscript has evidently been written with great care and attention to detail, but I have a few minor comments that the authors might like to consider –

Line Comment

27 Their phase noise

37 frequency-determining

56 time-keeping

79 were observed

92-101 Maybe these results/conclusions are premature & might fit better later in the paper?

185 In contrast

235 the change in frequency

392 amplitudes of (bottom to top) 9.5 [as the blues are not clearly distinguished]

406 For a resonator

411 such an approach

Otherwise, I recommend acceptance of the paper for publication in its present form.

Response to Reviewer 1

We are grateful to Reviewer 1 for the helpful comments and for recommending publication of our paper after his/her comments will have been addressed.

1. Reviewer 1 commented that "...I am not sure if the similar effect can be expected in the case when the noise was generated in the loop circuit because the fluctuation in VCO amplitude also increases the noise amplitude. If I correctly understand the result, the reduction of frequency deviation was observed because the injected noise is independent from the operation of upper stage of the circuit for example the pre-amplifier? I imagine the injected noise is also increased when the noise is amplified by the loop gain."

The effects of the injected noise on the vibrations is similar to that of thermal noise. It is indeed independent from the noise of the pre-amplifier. In our experiment, the injected noise is much stronger than the noise from the feedback loop even after noise from the pre-amplifier is amplified by the loop gain.

Even though our measurement only demonstrated phase stabilization by zero dispersion for the case of injected noise, the effect is expected to also apply to the case mentioned by Reviewer 1 when fluctuations in the VCO amplitude of the feedback loop dominates. We have added the following paragraph to discuss the effect of noise from the VCO amplitude and thank Reviewer 1 for bringing up this issue.

"Apart from the applied noise and thermal-mechanical noise, the feedback circuit also contributes to noise in the phase of oscillations. In particular, we consider the effects from the output of the voltage-controlled oscillator (VCO) that is used to drive the mechanical resonator. Fluctuations in the VCO output amplitude leads to amplitude noise in vibrations, but the phase dynamics are not directly affected if the resonator is linear. However, when the vibration amplitude is large, nonlinearity becomes strong and such amplitude noise could be converted into phase fluctuations. Designing and operating the resonator in the zero-dispersion regime reduces such conversion and is therefore expected to yield improved frequency stability, similar to the case of additive noise reported here. The experimental demonstration of such improvement, however, is outside the scope of this paper."

2. Reviewer 1 asked for an estimation of the effect in the case of Brownian motion.

We added the following paragraph to the discussions:

"The main effect of the injected noise in our experiment is that it increases the effective temperature of the resonator. If the injected noise is removed, the thermal noise at temperature of 4 K leads to random motion of the plate. In our experiment, such thermal motion is too small to be resolved due to noise in the detection circuit at room temperature.

To estimate the effect of zero-dispersion on preventing the conversion of amplitude fluctuations originating from thermal noise into phase fluctuations, we perform numerical simulations assuming that no additional noise is introduced by the feedback loop and the detection circuit. The optimal σ_ω in the zero-dispersion regime is found to be smaller than the conventional nonlinear regime by a factor of 4, comparable to the case measured in our experiment with injected noise.”

3. Reviewer 1 asked “Which is the major effect of applied noise, amplitude or phase? How is the effect of this method against intrinsic phase noise in mechanical resonator (say, mass fluctuation etc.)?”

The applied noise plays a role similar to thermal noise. In the linear regime, it leads to a distribution in the X-Y phase space that is isotropic (where X and Y are the amplitudes of the two quadratures, defined by Eq. 18). This isotropic distribution is centered at the origin of the phase space for no periodic driving. As the periodic driving amplitude increases, the distribution is displaced from the origin. The applied noise therefore leads to both amplitude and phase fluctuations (in the radial and tangential directions respectively). For a linear resonator, the phase fluctuations decrease with vibration amplitude (yielding the initial drop in Fig. 4d).

We added the following paragraph to elaborate on amplitude/phase noise, and explain that the effects of phase stability do not apply to intrinsic phase noise that is brought up by Reviewer 1:

“The improvement of frequency stability for resonators driven into self-sustained vibrations using feedback was demonstrated for applied noise that is additive. Such noise plays a role similar to thermal-mechanical noise that leads to both phase and amplitude fluctuations. At small vibration amplitudes, the direct contribution to phase fluctuations dominates. As the amplitude increases, conventional nonlinearity converts amplitude noise into phase noise. The effect of such a conversion is substantially reduced in the zero-dispersion regime with $d\omega/dE = 0$ so that amplitude fluctuations do not affect the phase stability even at large vibration amplitudes. If, instead, the phase noise is dominated by fluctuations in the eigenfrequency of the resonator¹³⁻¹⁶, operating the feedback loop in the regime of zero-dispersion does not yield improvement in phase stability because amplitude fluctuations are not involved.”

4. Reviewer 1 asked for comments on the upper limit of the zero-dispersion amplitude, which is chosen to be ~ 400 nm in this experiment.

We added the following paragraph on this upper limit:

“For the non-monotonic backbone curves, the optimal phase stability for self-sustained oscillations is achieved at the vibration amplitude A_{zd} at which $d\omega/dE = 0$. By eliminating the extra contributions to phase noise converted from amplitude noise, the phase stability becomes comparable to an oscillator undergoing purely linear vibrations at amplitude A_{zd} . It is well-known that for linear resonators, the frequency standard deviation for self-sustained vibrations is inversely proportional to the vibration amplitude⁵³. Therefore, the optimal frequency standard deviation for an oscillator in the zero dispersion regime is expected to follow a similar dependence on A_{zd} , i.e. $\sigma_{\omega,zd}^{(min)} \propto 1/A_{zd}$. Ideally, the phase stability can be improved by choosing A_{zd} to be as large as possible. The upper limit in our experiment is given by the gap g between the movable plate and the fixed electrode. In practice, operating close to this upper limit could lead to the plate touching the electrode when occasional large fluctuations occur. In Figs. 3 and 4, A_{zd} is ~ 400 nm, about a factor of 5 smaller than g . Assuming that A_{zd} can be increased to 80% of the upper limit, the ratio of the optimal frequency standard deviation in the zero dispersion regime to that in the conventional regime could reach ~ 12 , a factor of ~ 4 larger than in Fig. 4d. Demonstrating such best case scenario in our device, however, is not the main goal of this paper. Instead, we choose A_{zd} to be smaller than g to show the general features of zero dispersion phenomena. For example, to reveal the local minimum of the spectral width as a function of the noise intensity in Fig. 3d, it is necessary to go beyond the zero-dispersion regime using vibration amplitude that approaches the maximum displacement. Promising future research directions for phase stabilization using zero dispersion include new approaches to yield large A_{zd} , as well as the effect on the phase stability due to other phenomena that emerges when the vibration amplitude becomes large, such as nonlinear friction.”

The main reason for not choosing larger amplitudes is because we want to show clearly that zero dispersion effects go away when the amplitude is increased beyond 400 nm, for example, in Fig. 3d for noise intensity > 8000 pN² Hz⁻¹ and in the red curve in Fig. 4d for driving amplitude > 200 pN. Setting the zero dispersion amplitude at values larger than 400 nm by adjusting the dc voltage would indeed lead to better phase stability, but some of the important features in Fig. 3d and Fig. 4d will not show up clearly. For the second part of the paper, the data on self-sustained vibrations with a feedback loop was taken with a different cool-down. We chose V_{dc} so that the zero-dispersion amplitude is close to that in the first part of the paper.

Response to Reviewer 3.

We are grateful to Reviewer 3 for finding our paper to be interesting and novel, and for recommending acceptance for publication.

We have incorporated all of the suggestions of reviewer 3, as marked in the revised manuscript, except two items that we describe below.

1. Line 235. “the change in frequency”

We agree with Reviewer 3 that the original text of “frequency vs. the square vibration amplitude” is unclear. While the suggestion of Reviewer 3 clarifies the sentence, we think it is better to completely eliminate the ambiguity by using “the shift in ω_{peak} relative to the natural frequency vs the squared vibration amplitude.”

2. Line 92-101 “Maybe these results/conclusions are premature & might fit better later in the paper?”

We understand that Reviewer 3 might find the sentences on results/conclusions premature because a thorough understanding would require reading the rest of the paper. Nevertheless, we believe that it is necessary to provide a brief formulation of the main results in the introduction. In the earlier parts of this paragraph, we stated that “Our system opens the possibility for the investigation of a wealth of phenomena in zero-dispersion systems...”. We hope to convey to the readers that out of the aforementioned “a wealth of phenomena”, we focus on two of them: spectral narrowing and frequency stabilization with active feedback.

We have modified part of the text to make it more clear and concise (red represent changes):

“We demonstrate that in the zero-dispersion regime, the **peak of the fluctuation spectrum** undergoes substantial narrowing as the noise intensity is increased, **in good agreement with theory**. We also set our devices into self-sustained oscillations with active feedback. The **standard deviation of the oscillation frequency** at the optimal vibration amplitude is demonstrated to be a factor of ~ 3 smaller in the zero-dispersion regime compared to the **minimal noise in the conventional nonlinear regime. ...**”.

REVIEWERS' COMMENTS:

Reviewer #1 (Remarks to the Author):

I recommend the publication since the authors revised the manuscript reflecting all my comments.

Reviewer #3 (Remarks to the Author):

In my opinion, the original submission was already suitable for publication, subject to minor corrections. These have now been implemented, and further improvements made in the light of the report from Reviewer 1.

So I recommend that the manuscript should now be accepted for publication in Nature Communications.

REVIEWERS' COMMENTS:

Reviewer #1 (Remarks to the Author):

I recommend the publication since the authors revised the manuscript reflecting all my comments.

Response: We thank Reviewer 1 for recommending publication of our paper.

Reviewer #3 (Remarks to the Author):

In my opinion, the original submission was already suitable for publication, subject to minor corrections. These have now been implemented, and further improvements made in the light of the report from Reviewer 1.

So I recommend that the manuscript should now be accepted for publication in Nature Communications.

Response: We thank Reviewer 3 for recommending publication of our paper.